# Multiple Mechanisms to Strengthen the Ability of YOLOv5s for Real-Time Identification of Vehicle Type

Qiang Luo [1,2,3], Junfan Wang [1,3], Mingyu Gao [1,3,*], Zhiwei He [1,3], Yuxiang Yang [1,3] and Hongtao Zhou [4]

1    School of Electronics and Information, Hangzhou Dianzi University, Hangzhou 310018, China
2    School of Communication and Electronics, Jiangxi Science and Technology Normal University, Nanchang 330038, China
3    Zhejiang Provincial Key Lab of Equipment Electronics, Hangzhou 310018, China
4    Zhejiang Leapmotor Technology Co., Ltd., Hangzhou 310018, China
*    Correspondence: mackgao@hdu.edu.cn; Tel.: +86-133-8651-0408

**Abstract:** Identifying the type of vehicle on the road is a challenging task, especially in the natural environment with all its complexities, such that the traditional architecture for object detection requires an excessively large amount of computation. Such lightweight networks as MobileNet are fast but cannot satisfy the performance-related requirements of this task. Improving the detection-related performance of small networks is, thus, an outstanding challenge. In this paper, we use YOLOv5s as the backbone network to propose a large-scale convolutional fusion module called the ghost cross-stage partial network (G_CSP), which can integrate large-scale information from different feature maps to identify vehicles on the road. We use the convolutional triplet attention network (C_TA) module to extract attention-based information from different dimensions. We also optimize the original spatial pyramid pooling fast (SPPF) module and use the dilated convolution to increase the capability of the network to extract information. The optimized module is called the DSPPF. The results of extensive experiments on the bdd100K, VOC2012 + 2007, and VOC2019 datasets showed that the improved YOLOv5s network performs well and can be used on mobile devices in real time.

**Keywords:** vehicle type detection; object detection; G_CSP; C_TA; DSPPF





## 1. Introduction

Object detection [1] is a basic task in computer vision that has attracted growing research interest in recent years. Designing a valid neural network structure based on the CNN is the main means of object detection in natural scenes. For example, object detection methods are used to efficiently identify the type of vehicle and its license plate number in the context of intelligent transportation. One-stage and two-stage methods are the major frameworks used for object detection. When predicting the classes and locations of objects, the one-stage method can be used to directly extract features from the feature map. The YOLO series is an example of this [2,3]. This study uses logistic regression to predict the objectiveness score of each bounding box (bbox) [4]. In the implementation of the algorithm, if a bounding box overlaps with the ground truth object more than any other bounding box a priori, its value is set to one. When the prior value of the bounding box is not optimal, the algorithm ignores the predicted value even if it overlaps with the real ground truth value of the object beyond a certain threshold. Gaussian YOLOv3 [4] can not only improve the accuracy of detection of the algorithm, but can also support its real-time operation. It involves redesigning the loss function and using Gaussian parameters to model the bbox of YOLOv3.

A considerable amount of research has been reported on the one-stage object detection network. The SSD [5] uses the feature pyramid network to extract feature-related information. It can improve the capability of the algorithm to detect large and small objects by extracting feature maps at different scales. The shape of the detected object can be better

matched by adjusting the prediction box. CenterNet [6] is an object detection framework from the anchor-free series that is responsible only for predicting the center of the object for the detection task. This detector can, thus, efficiently explore the visual pattern in each clipping area of the whole image at the lowest cost. Each object is detected as a triple of key points, instead of a pair, to improve the accuracy and recall of the detection network. While these networks achieve good performance on object detection tasks, the number of parameters of their models is too large, and they require considerable hardware resources to run on mobile devices.

It is often necessary to detect and identify the types of vehicles on the road in the context of intelligent transportation before detecting whether they have violated any traffic law. The traditional methods of detection often suffer from such defects as inaccurate identification, a high cost, and fixed location of detection. Convolution neural networks (CNNs) have been used as a superior alternative to identify the type of vehicle. Zhao et al. [7] strengthened the feature extraction ability of the down-sampling algorithm by adding to it an attention mechanism and a feature pyramid model based on the YOLOv4 object detection algorithm, but did not test their model on an embedded mobile platform. Khalifa et al. [8] used the YOLOv5s model and the k-means clustering algorithm to improve the detection of the type of vehicle under different lighting conditions, and achieved especially good results in cases involving low illumination. However, their method can only detect vehicles in images acquired from fixed monitoring equipment and not in images obtained by on-board monitoring equipment. Park et al. [9] developed an algorithmic framework that can simultaneously detect the type of vehicle and its license plate information using the YOLOV4 algorithm, and created a dataset of types of vehicles. However, their algorithm has stringent requirements on the performance of the equipment used, and cannot satisfy the real-time requirements of Jetson AGX. Li et al. [10] used images obtained from unmanned aerial vehicles (UAVs) for vehicle identification and proposed a data enhancement method to solve the problem of class imbalance. They separated the vehicle from the background information in a given image through semantic segmentation, and randomly replaced instances in the over-represented class with those in the under-represented class. This algorithm does not require additional high-quality segmentation masks. Li et al. [11] proposed a multi-view vehicle detection system that uses part models to solve the problem of partial occlusion and the differences between vehicle types. The part model proposed was visual and could be replaced at any time. Although these algorithms can detect vehicles on the road, the file of the model is too large to be deployed to the on-board mobile terminal, and their inability to extract features related to the vehicle leads to a poor overall accuracy of detection. The attention mechanism can increase this capability of the network without increasing the amount of required computation.

Attention mechanisms are commonly used in object detection tasks as they can extract useful information from feature maps to improve detection performance. One example is the SE [12] network in which the attention-related information for each channel is obtained by using a global pooling layer. After it passes through a fully connected layer and the sigmoid operation has been applied to it, it is multiplied by the original feature map to strengthen attention-related information. CBAM [13] contains an attention mechanism composed of spatial and channel attention, in which channel attention is used to extract the channel-related information of the feature map. The mechanism of spatial attention involves extracting information on the global maximum pooling and global average pooling of the feature map. This attention mechanism can help improve the performance of the object detector by extracting attention-related information. The TA [14] has a triple attention network that can explain cross-dimensional informational interaction. It consists of three branches, and each extracts the attention-related information of the spatial and channel dimensions from the feature map. The efficiency of this method allows the network to fully extract information in the feature map. FcaNet [15] contains an attention mechanism based on the customized discrete cosine transform (DCT). However, this operation only extracts information in the channel or the spatial dimension, and ignores feature-related information

from other dimensions. The TA uses the rotation operation to obtain 3D attention-related information from the feature map but integrates the information of each dimension by using the summation method, where this renders the network unable to highlight important information in the feature map. M. Hamed Mozaffari et al. [16] simulated the peripheral ability of human eyes. A novel convolution module was proposed, which combined standard and generalized convolution to extract features. By using this module in encoder decoder configuration, good results were achieved in several common datasets.

Although the above studies have achieved good results in the detection of vehicles, they still cannot meet the requirements of real-time deployment on mobile platforms. It is important to develop such a vehicle detection algorithm. To this end, we propose a modified YOLOv5s network called YOLOv5s+, which can enhance the capability of the network for feature extraction by introducing a large-scale convolution layer, optimizing the rotating attention module, and using dilated convolution instead of the maximum pooling operation. This not only improves performance in terms of detection of the type of vehicle, but also satisfies the requirement of real-time operation on mobile devices. The main contributions of this work are as follows:

- We have devised a G-CSP module based on YOLOv5s' cross-stage partial network (CSPNet). It can efficiently extract information from feature maps by using a large-scale separable convolution network to improve the accuracy of detection of types of vehicles.
- We have developed a C_TA module based on the TA module. We used concatenation and convolution operations to fuse multi-dimensional attention-related information, such that the network can assign different attention scores to different dimensions of this information to improve detection performance.
- We propose the DSPPF module based on YOLOv5s' SPPF module that replaces the maximum pooling operation in the original SPPF module with a dilated convolution operation. It can increase the perceptual field of the network for detecting targets with only a slight increase in the requisite computational effort.

## 2. Related Work

Attention mechanisms have been extensively used in many areas of computer vision, such as image classification [17], object detection [18], instance segmentation [19], semantic segmentation [20], scene parsing, and action localization. Channel attention and spatial attention are the two most widely used attention mechanisms. Recent research has shown that significant improvements in performance can be achieved by employing channel attention, spatial attention, or both. The SE module is the most commonly used method of channel attention. This study analyzes the channel-related relationship in the network structure, and uses the global average pooling operation to extract channel-related information from the feature layer to enable the network to pay more attention to the relationship between feature channels. To satisfy the above requirements, we propose a channel-based attention mechanism that allows the network to optimize the features when performing the detection task. Through this mechanism, the neural network can use global information to strengthen the useful information that it contains and suppress less useful features. The CBAM uses two independent dimensions of channel and space to obtain attention-related features in turn, and multiplies the attention map by the input feature map for adaptive feature refinement. It can be easily embedded into any CNN architecture. Besides the cost of end-to-end training with the basic CNN, the computations of the system are negligibly small. The TA improves the performance of attention mechanisms by using different dimensions. We examined a lightweight but effective attention mechanism, and propose triplet attention. This can be used to capture cross-dimensional interaction in the feature map through a three-branch structure to calculate the attention weights. For the input tensor, triple attention interacts with information on different dimensions through the rotation operation and residual transformation, and the amount of calculation of this method is sufficiently small to be ignored. It can also be easily applied to other network structures.

The efficiency of the mobile model is often considered in the context of its use. Mo-bileNetV1 [21] provides the idea of designing a neural network with two characteristics: a small number of parameters and a small model, allowing the network to run on a mobile device. The network replaces the original convolution with a deep, separable convolution network and a $1 \times 1$ convolution operation to improve the speed of detection of the net-work. MobileNetV3 [22] contains a new convolution unit composed of a depth convolution and a point convolution. A large convolution kernel is used to approximate the original convolution layer, and its detection-related performance is mostly equivalent to that of the original convolution. Ghostnet [23] introduces a ghost module that uses few param-eters while generating the same size of feature maps. The network divides the ordinary convolution operation into two steps. In the first step, the original convolution operation is used to output a certain number of feature maps that contain information on different layers of the original feature map. In the second part, the deep, separable convolution network is used to generate another part of the feature map. The two parts of the map are output as a new feature map following the concatenation operation. These operations can be used to generate the same feature map as the original convolution operation with a considerably smaller number of parameters. Some researchers use multi-sensor infor-mation fusion to improve the algorithm's vehicle detection and traveling area recognition in road scenes. Zeng X et al. [24] found that the feature fusion using deep convolution network did not consider the matching degree between different features. Therefore, they proposed fusion filter as feature maps matching technology to solve the problem of feature mismatch. At the same time, the layer sharing technology in the deep layer proposed by them can achieve better accuracy with less computational overhead. The method proposed by them enables DCNN to learn corresponding feature maps with similar features and complementary visual backgrounds from different modes to obtain better accuracy. Sorosh Bateni et al. [25] found that different memory management methods in real applications would affect the performance of the platform. They proposed a runtime scheduler, which can reduce the memory pressure during system operation, and achieved good results on NVIDIA Jetson TX2, drive PX2, and Xavier AgX platforms. Xudong Dong et al. [26] used the yolov5 network framework to complete the vehicle type detection task. They used the ghost module to reduce floating-point operations (FLOPs) of the model, and used attention mechanism and CIOU_ Loss to improve the detection performance of the algorithm.

## 3. Proposed Approach

In this section, we briefly introduce the framework of YOLOv5, as shown in Figure 1. This network has a small model, high accuracy of detection, and can easily be deployed. We subsequently revisit CSPNet and analytically diagnose the efficiency of the G-CSP module. We also propose our C-TA module, in which we use the convolution operation to replace the original addition to integrate attention-related information from different dimensions. Finally, we introduce the DSPPF module, which is extended by the SPPF module in YOLOv5s. We use dilated convolutions of different convolution cores to replace the original maximum pooling operation and improve the capability of the network for feature extraction.

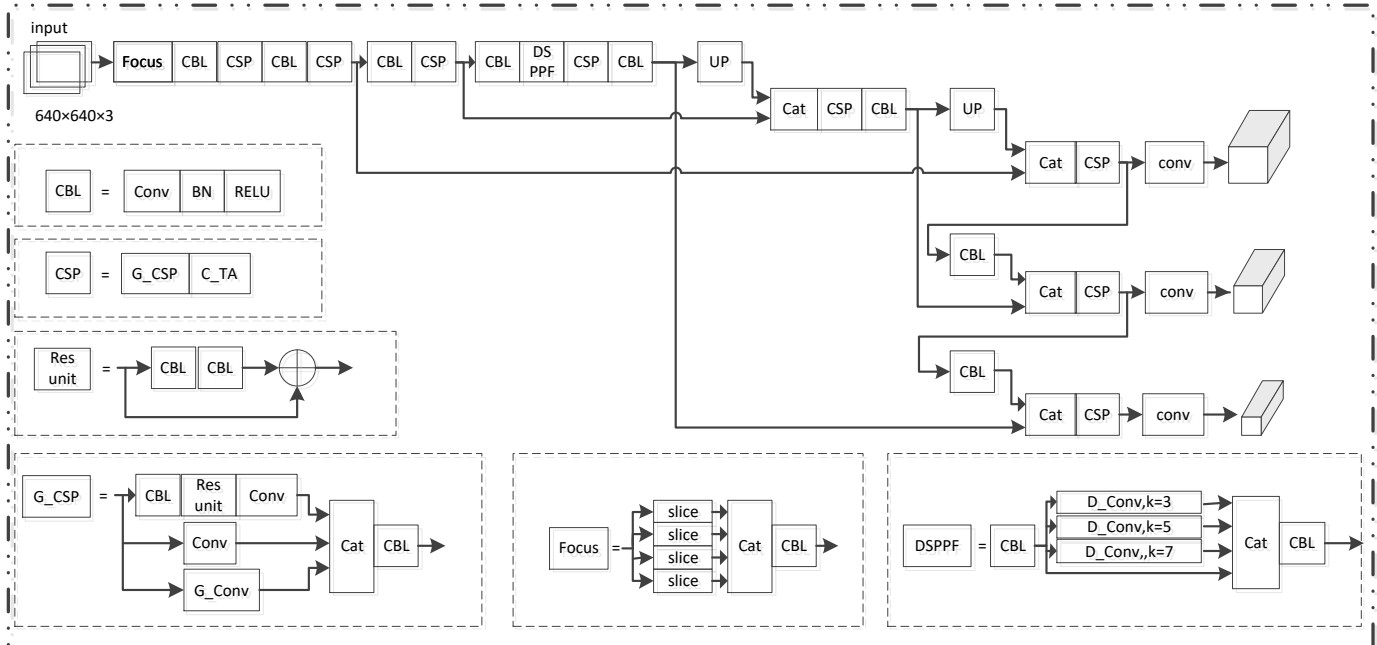

**Figure 1.** YOLOv5s module. Conv denotes general convolutions operation. BN denotes Batch Normalization. RELU denotes Relu activation function. Res unit denotes residual structure. ⊕ denotes element-wise addition. CSP denotes the cross-stage partial network. G_CSP denotes the ghost cross-stage partial network. C_TA denotes convolutional triplet attention network. DSPPF denotes the dilated convolution with spatial pyramid pooling-fast module. Cat denotes the characteristic concatenation operation. Focus denotes slice operation.

### 3.1. YOLOv5s Module

YOLOv5 [27] is the most popular object detection network in the area. Some of its design has been inspired by that of YOLOv4 [28]. This network has a high speed of detection and accuracy, and its model is very small, such that it is suitable for deployment on mobile devices. PANet [29] is also used in this network to improve information fusion at different scales and optimize the performance of the detector. We use the YOLOv5s module to identify the type of vehicle. It is a scaled-down version of YOLOv5 that still performs well on object detection tasks. We have optimized the CSP module in this framework, as explained in Sections 3.2 and 3.3.

### 3.2. G_CSP Module

From the perspective of the design of the network structure, CSPNet solves the problem of a large amount of calculations in the reasoning process encountered in past work, as shown in Figure 2. It uses two methods to obtain an output feature map; one part uses a $1 \times 1$ convolution, and the other part uses a $3 \times 3$ convolution and the ResNet [30] network. The results of the two convolutions are fused to generate the output feature map of this module.

However, this CSPNet convolution module has only $1 \times 1$ and $3 \times 3$ convolution kernels, and cannot extract large-scale information from the feature map. We use the large-scale depth-separable convolutional network to extract such information, and call this the G_CSPNet module, as shown in Figure 3. G_conv is a large-scale convolutional network. This part of the feature map is fused with the feature map of the original CSPNet module to obtain a feature map containing large-scale information.

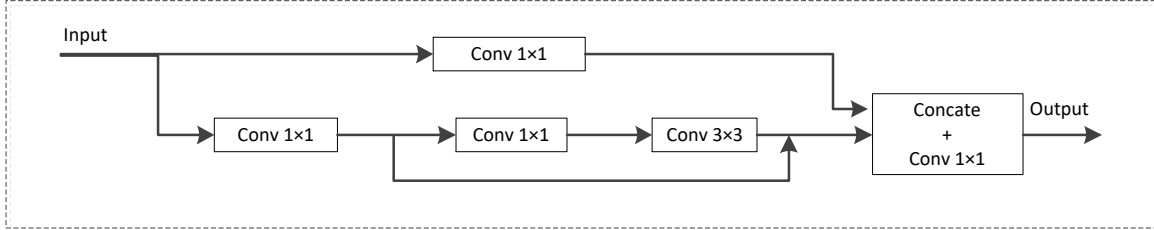

**Figure 2.** CSPNet module. Conv 1 × 1 denotes general convolution operations with kernel = 1. Conv 3 × 3 denotes general convolution operations with kernel = 3. Concate denotes the characteristic concatenation operation.

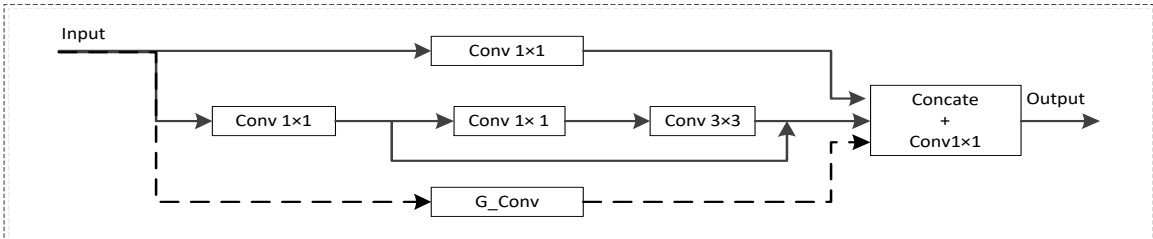

**Figure 3.** G-CSPNet module. G_Conv denotes large-scale convolution operation.

The G_conv module is composed of a CNN, as shown in Figure 4. The 1 × 1 convolution kernel is first used to change the size of the feature map from C × H × W to 1 × H × W, and a 9 × 9 convolutional network is then used to extract large-scale information from the feature map by using it in a cascade with the previous 1 × 1 convolutional network, called G_conv. This improves the performance of the detector by extracting large-scale information from the feature map while adding few parameters.

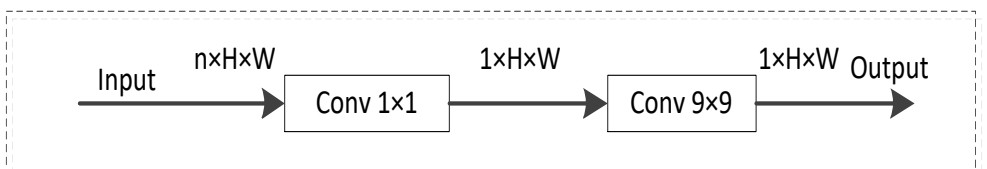

**Figure 4.** G_conv module. Conv 1 × 1 denotes general convolution operations with kernel = 1. Conv 9 × 9 denotes general convolution operations with kernel = 9.

### 3.3. C_TA Module

Triplet attention (TA) is an efficient, cross-dimensional, and interactive attention mechanism. It consists of three branches, each of which is responsible for extracting attention-related information from the input spatial and channel dimensions. The features of each branch are then fused through weight addition. After applying the attention weight to the arranged input tensor, the input tensor is permuted into the original input. This enables the network to extract information from the feature map along different dimensions by adding the permutation operation, as shown in Figure 5a.

There is a prominent defect in the default triplet attention mechanism, whereby adding the attention-related information equally from the three dimensions renders the detector unable to filter useful information from the feature map. To solve this problem, we have improved the module for the attention mechanism. As shown in Figure 5b, the original average addition is changed to weight addition by using the 1 × 1 convolution kernel so that the detector can extract the important information from the feature map according to the needs of the detection task while ignoring useless information.

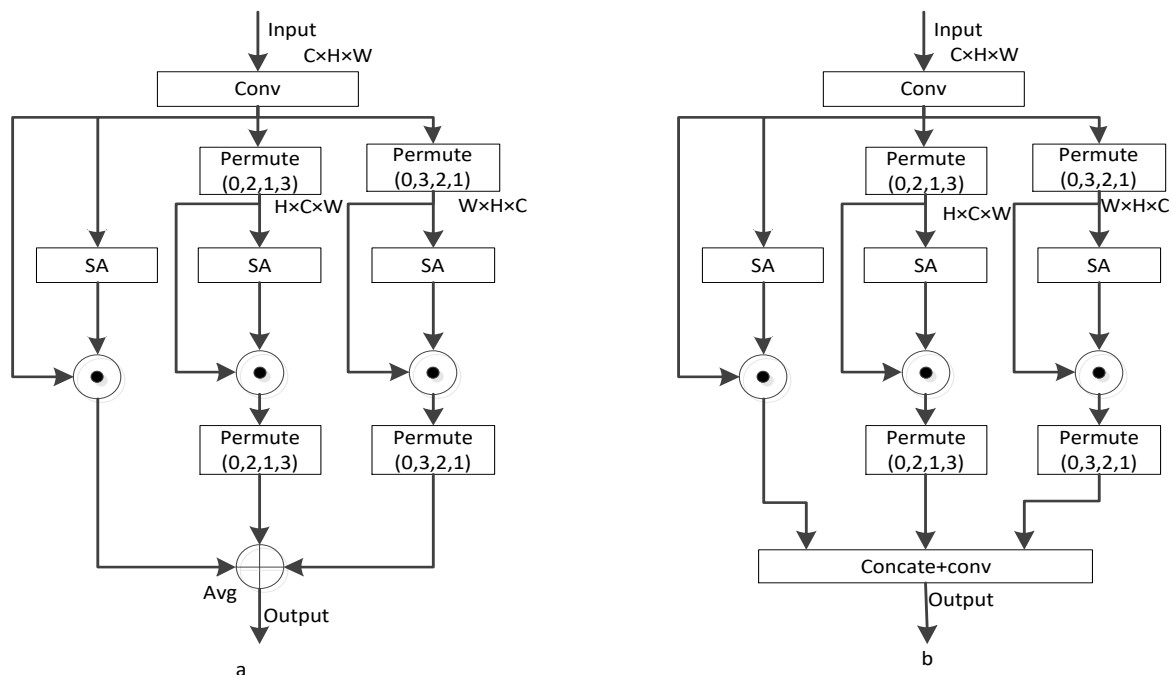

**Figure 5.** Comparisons of different triplet attention modules: (**a**) Default triplet attention (TA) module. (**b**) Concatenation-based and convolutional triplet attention module (C_TA) (ours). The feature maps are denoted by feature dimensions, e.g., C × H × W denotes a feature map with C channels, height H, and width W. Conv denotes general convolution operation. Permute denotes rotation operation for rotating dimensions. Avg denotes mean operation. ⊙ denotes element-wise multiplication and ⊕ denotes element-wise addition. SA denotes spatial attention.

### 3.4. DSPPF Module

The SPPF module is used as the backbone of the framework of YOLOv5s. It is composed of maximum pooling operations of different sizes to further improve the capability of the network for feature extraction. In addition, maximum pooling operations of different sizes are used in the network to extract useful information. The three pooled outputs are concatenated with the original feature map as the overall output of the module. The SPPF module as shown in Figure 6a.

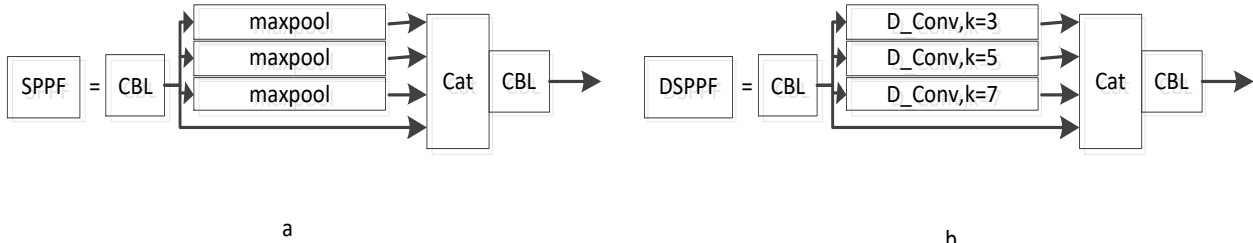

**Figure 6.** Comparisons of different SPPF and DSPPF modules: (**a**) Default SPPF module. (**b**) DSPPF module (ours). The feature maps are denoted by feature dimensions, e.g., D_conv denotes dilated convolutions, and k = 3, 5, and 7 are the mean sizes of the convolution kernel. maxpool denotes maximum pooling operation. CBL denotes general convolution operation, batch normalization, and Relu activation function. "Cat" represents the characteristic concatenation operation.

The traditional SPPF network has a disadvantage in that it extracts only the maximum value of each element while ignoring the other elements. This leads to a loss of useful information in the feature map. We, thus, designed a dilated convolution, called the DSPPF module, to replace the maximum pooling operation, as shown in Figure 6b. The dilated convolution is similar to the original convolution operation. Different convolution cores are

set to obtain information from different receptive fields, but it has a larger receptive field than the original convolution operation. Although this operation increases the amount of computation compared with maximum pooling, it can extract more useful feature-related information to enhance the detection performance of the algorithm.

## 4. Experiments and Results

### 4.1. Experiment Environment

In this section, we mainly show our work through detailed experimental steps and data analyses. In several groups of comparative experiments, we compared the improved attention mechanism with ones reported in the literature. We verified our method on several challenging computer vision tasks, such as object detection on a car dataset VOC2019 [30] that we created, PASCAL VOC [31], bdd100k datasets [32], as well as. We used the YOLOv5s network as the basic network. This object detection network, with a size of only 14.5M, is suitable for running object detection tasks on embedded devices. Our improved network is called YOLOv5s+, and we compared its performance with that of YOLOv5s. To further verify our method, we conducted ablation experiments on the G_ CSP and C_TA modules to verify their effectiveness in extracting feature-related information from the feature map. Our experiments were performed on a Linux Ubuntu 18.04 with $8 \times$ TITAN XP and 12 GB of memory. The parameters of our algorithm were set as follows: The input image size was set to $640 \times 640$ and mosaic was used for data augmentation. The initial learning rate was set to 0.01 and the weight_decay was set to 0.0005. Momentum was set to 0.937. The training epoch was set to 300. In all experiments, the same super parameters were used. We trained on bdd100K, VOC2007 + 2012, and VOC2019 vehicle type datasets, while one GPU was used for training. The performance in terms of frames per second (FPS) was tested on embedded development boards, such as Jetson Xavier NX.

We appraised the performance of our algorithm along the following dimensions: model parameter quantity (Params (M)), floating point operations per second (flops, 1 Gflops = $10^9$ flops), model weight file size (weights (MB)), mean average precision IOU = 50% (mAP50), mean average precision IOU = 50%:0.05:95% (mAP50_95), and frames per second (FPS). The results are detailed below.

### 4.2. Quantitative Evaluation

#### 4.2.1. bdd100k

The bdd100k dataset was formulated and is sponsored by the Berkeley DeepDrive industry alliance. This dataset is the largest, most complex, and most diverse open dataset of automatic driving videos. It contains labeled information on vehicles, lanes, and pedestrians on urban roads. Many studies on object detection have used this dataset for performance verification. It contains 10 categories of GT box labels and a large number of objects from natural scenes that pose a daunting challenge to the detection task. We improved the YOLOv5s object detection framework to increase map_50 by 1.9% and map50_95 by 1.6%. Table 1 shows that our algorithm improved detection performance even on this large-scale and multi-scene dataset.

**Table 1.** Test of improved YOLOv5s on bdd100k dataset.

| Model | Params (M) | Gflops | Weights (MB) | map50 | map50_95 |
|---|---|---|---|---|---|
| YOLOv5s | 7 M | 15.9 | 14.1 | 0.493 | 0.260 |
| YOLOv5s+ (ours) | 10 M | 22.7 | 20.1 | 0.512 | 0.276 |

#### 4.2.2. VOC2019

We created the VOC2019. It can be used for detecting the types of vehicles. We captured the images of vehicles in Wenzhou, China, by using MV-CA050-10GM/GC digital cameras with 5,000,000 pixels. This dataset contained scenes from roads. Table 2 shows that map50 in this dataset was improved by 2.1%.

**Table 2.** Test of improved YOLOv5s on VOC2019 car dataset.

| Model | Params (M) | Gflops | Weights (MB) | map50 | map50_95 |
|---|---|---|---|---|---|
| YOLOv5s | 7 M | 15.9 | 14.5 | 0.741 | 0.542 |
| YOLOv5s+ (ours) | 10 M | 22.7 | 20.25 | 0.762 | 0.553 |

### 4.2.3. VOC2007 + 2012

We also tested our framework on the VOC2007 + 2012 datasets, which are general datasets for target detection tasks. It contains 20 classes. We used images from VOC2012 and the training set of VOC2007 as the training data, and the test set of VOC2007 as the test images. Table 3 shows that our algorithm improved detection performance on the VOC2007 + 2012 datasets.

**Table 3.** Test of improved YOLOv5s on the VOC2007 + 2012 dataset.

| Model | Params (M) | Gflops | Weights (MB) | map50 | map50_95 |
|---|---|---|---|---|---|
| YOLOv5s | 7 M | 15.9 | 14.5 | 0.796 | 0.550 |
| YOLOv5s+ (ours) | 10 M | 22.7 | 20.7 | 0.825 | 0.589 |

### 4.3. Ablation Study

We examined the impact of using different bag-of-freebies (BOF) detectors in the object detection network on its accuracy of training. To study the influence of different strategies on the performance of the detector, we significantly expanded the BOF list, as shown in Table 4. Case 1 involved using YOLOv5s as the basic network, case 2 used G-CSP module replacement Cross-Stage Partial Network (CSPNet) in YOLOv5s, case 3 added TA network behind CSPNet, case 4 added C_TA network behind CSPNet, case 5 used G_CSP to replace CSPNet and add C_TA module in YOLOv5s network, and case 6 used DSPPF to replace the SPPF module in YOLOv5s network.

**Table 4.** Ablation studies on bag-of-freebies using different components (YOLOv5s, 640 × 640).

| Case | Our Basic | G_CSP | Default_TA | C_TA | DSPPF | Params (M) | map50 | map50_95 | FPS |
|---|---|---|---|---|---|---|---|---|---|
| 1 | √ | | | | | 7 | 0.796 | 0.550 | 60 |
| 2 | √ | √ | | | | 7.8 | 0.807 | 0.562 | 58 |
| 3 | √ | | √ | | | 9.3 | 0.801 | 0.555 | 42 |
| 4 | √ | | | √ | | 9.3 | 0.810 | 0.565 | 40 |
| 5 | √ | √ | | √ | | 10 | 0.816 | 0.582 | 37 |
| 6 | √ | √ | | √ | √ | 10.1 | 0.825 | 0.589 | 35 |

Figures 7–9 show the results of detection of our YOLOv5s network on the PASCAL VOC, bdd100k, and VOC2019 datasets. As shown in the figures, the algorithm directly outputs the category, probability, and coordinate information of the detection target. Our research team developed a small object detector, such that the entire model could run in an embedded mobile system for the real-time detection of vehicles on the road. Our algorithm correctly identified conventional vehicles on the road.

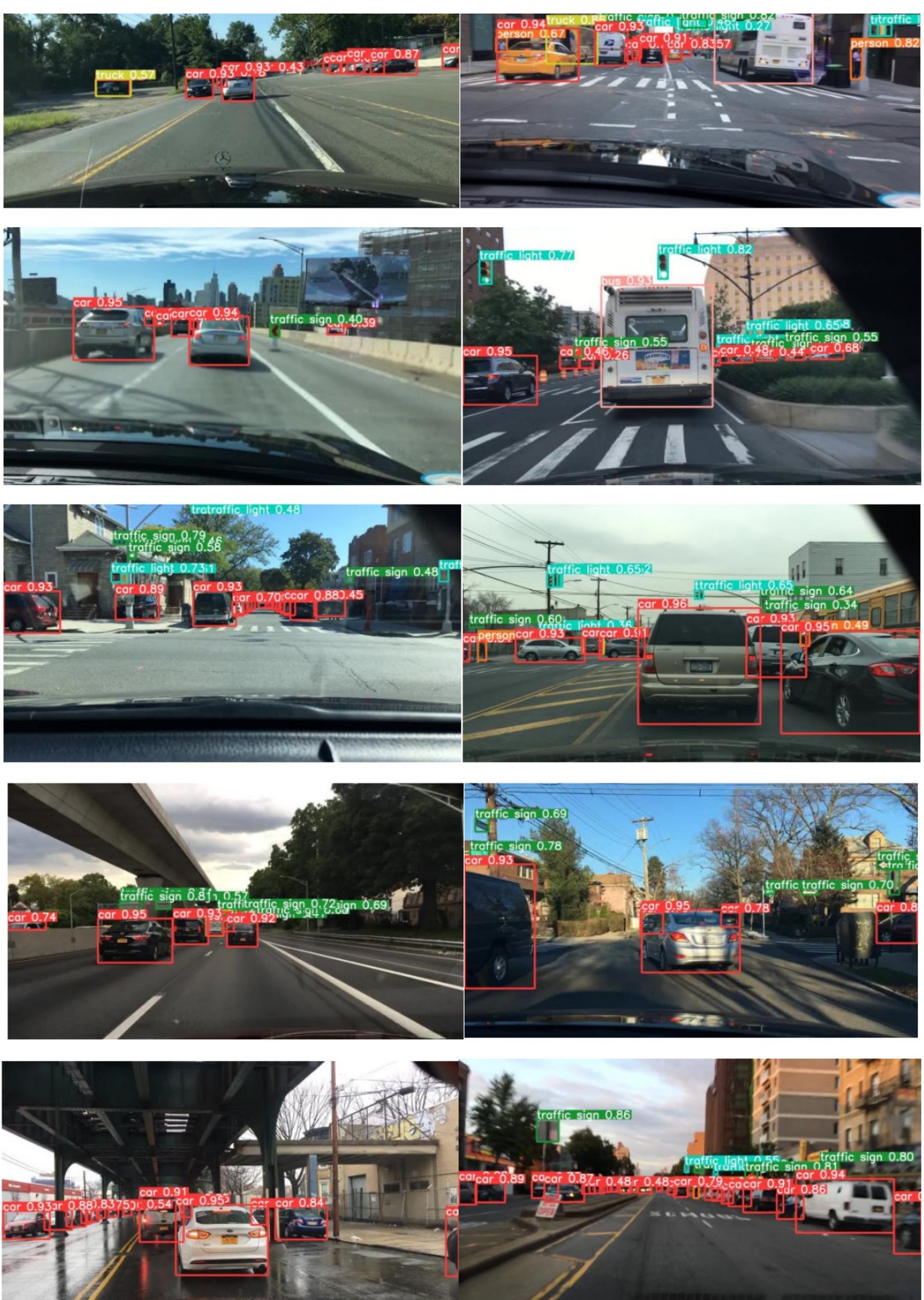

**Figure 7.** Results on the bdd100k dataset.

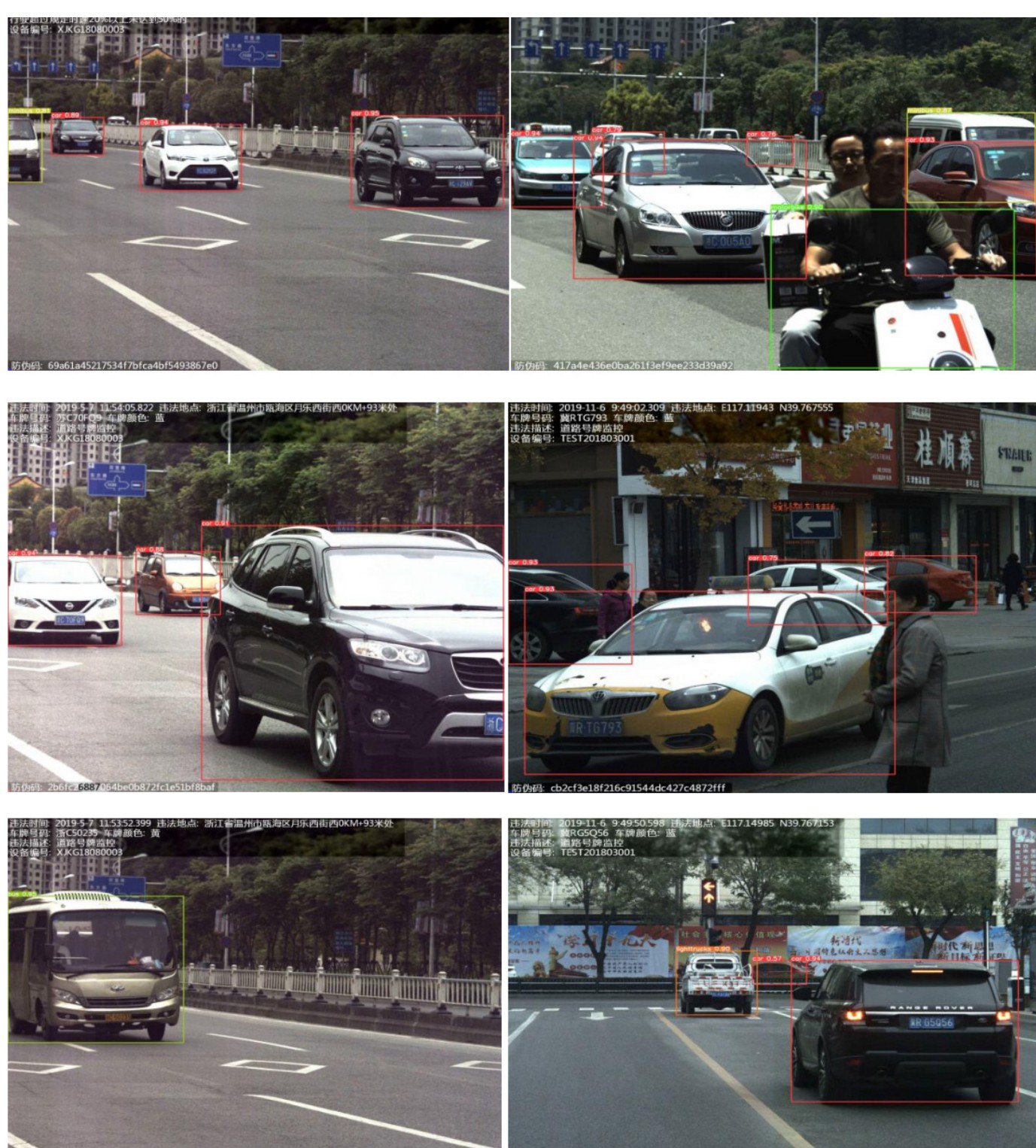

**Figure 8.** Results on the VOC2019 dataset.

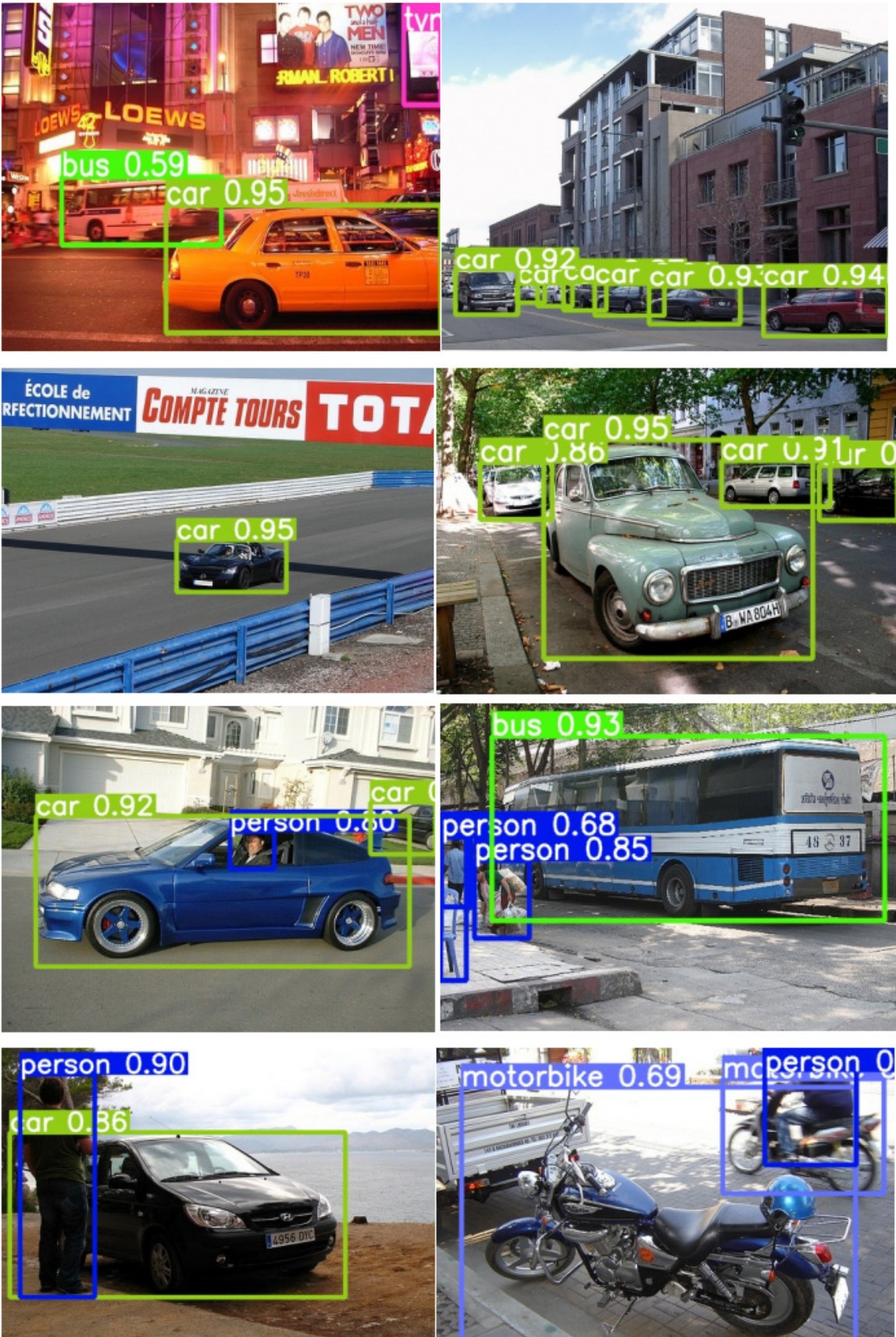

**Figure 9.** Results on the VOC2007 + 2012 dataset.

## 5. Conclusions

At present, many researchers use convolutional neural networks to complete specific visual tasks [33–35], and use attention mechanism [36], large-scale convolution and other strategies [37,38] to improve the performance of algorithms. In this paper, we mainly completed the task of detecting vehicle types in the road scene. In order to meet the real-time operation of the algorithm in embedded devices, we used yolov5s as the main detection framework, and improved the detection performance of the algorithm by introducing a large-scale convolution function, a multi-dimensional attention mechanism, and increasing the correlation between adjacent pixels. However, in the experiment, we found that the above strategy can increase the detection accuracy of the algorithm, but it increases the detection time of the algorithm. Therefore, in the next work, we will study more efficient vehicle type detection algorithms, further reducing the detection time of the algorithm.

**Author Contributions:** Conceptualization, Q.L. and M.G.; methodology, Q.L.; software, Q.L. and J.W.; validation, Q.L. and Y.Y.; formal analysis, Z.H.; investigation, H.Z.; resources, M.G.; data curation, Q.L. and J.W.; writing—original draft preparation, Q.L. and J.W.; writing—review and editing, Y.Y.; visualization, Y.Y.; supervision, M.G.; project administration, M.G.; funding acquisition, M.G. All authors have read and agreed to the published version of the manuscript.

**Funding:** This work was supported by the Zhejiang Provincial Major Research and Development Project of China, under Grant 2022C01062, and by the Zhejiang Provincial Key Lab of Equipment Electronics.

**Acknowledgments:** We thank Leapmotor Technology Co., Ltd. Of Zhejiang for providing support for our tests.

**Conflicts of Interest:** The authors declare no conflict of interest.

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
