# Peer review of "Multiple Mechanisms to Strengthen the Ability of YOLOv5s for Real-Time Identification of Vehicle Type"

_electronics, doi:10.3390/electronics11162586_

Round 1
Reviewer 1 Report
This paper targets identifying the vehicle types on the road and proposes a large-scale convolutional fusion module called ghost cross-stage partial network (G_CSP) based on YOLOv5s backbone to integrate large-scale information from different feature maps to identify vehicles on the road. Also, the design involves other techniques, including using convolutional triplet attention network (C_TA), optimizing spatial pyramid pooling fast (SPPF) module, and adopting dilated convolution, The final evaluations demonstrate the performance of the proposed design.
Overall, the paper targets an interesting and meaningful problem and proposes a large-scale network to address the problem. The designs are detailed, and the evaluations demonstrate the performance improvement of the design.
However, there are several concerns, and it is advised that the authors can address them.
1. It is advised to add the model training details.
2. It appears that the size of YOLOv5s is 14MB instead of 7MGB (https://github.com/ultralytics/yolov5/issues/3416). Meanwhile, as Tables 1 - 4 show, with the backbone YOLOv5s integrated with several other modules, the proposed model only consumes 10M memory, I wondered how it is achieved. It is advised to add more profiling details here.
3. Regarding Athe blation Study, can the authors add more details about the profiling? What is the batch size and what profiling tools are used? How to achieve the 60 FPS of the baseline? Meanwhile, as the design targets real-time vehicle types identification, if the FPS drops to around 30 FPS with more modules integrated with the baseline, I wondered whether the designed networks still work for the target applications.
4. Can the authors add more explanations to Table 1? What do the results of row2 and row 4 indicate? What is the difference between them? It is confusing.
5. Can the authors report the TOP1 and TOP-5 accuracy?
6. Please discuss related works 10.1109/RTAS48715.2020.00007, 10.1016/j.engappai.2022.104914, arXiv:2202.11231.
7. Can the authors revise Fig. 7? The font is too small and the figures are blurred. The Figures deliver very limited information.
8. Please pay attention to the typos in the abstract: conv triplet attention -> convolutional triplet attention, pooling - fast -> pooling-fast
9. What do the 2nd and 3rd paragraphs indicate on page 8? I wondered whether they should be deleted.
Author Response
Thank you very much for giving us an opportunity to revise our manuscript. We have perfected our manuscript according to your modification opinions. Please see the attachment for details. Thank you again for your valuable suggestions for our paper

Reviewer 2 Report
The authors proposed a modification for YOLO5 for single-stage object detection. Their idea is to add modules and logistic regression for this idea. The contribution is not significant but qualitative experimental results are interesting.
The article has many problems that should be fixed before considering for publication.
Main suggestions for round one of the review:
1- Figures have poorly been explained. For example! figure 1, where is the explanation of acronyms? Readers can't get the idea from looking at your figures. what is the specification of each box? Also for other figures, Conv1*1 is not a standard way of explanation for figures, you need to define them.
2- Try to explain all acronyms as possible in context and figures.
3- Experimental figures are so poor in quality.
4- the same problem with tables. The caption of the table should explain all details about the table.
5- section 2, there are also other types of attention for semantic segmentation like: Mozaffari, M. Hamed, and Won-Sook Lee. "Semantic Segmentation with Peripheral Vision." International Symposium on Visual Computing. Springer, Cham, 2020.
6- In general, section 1 is too long. I suggest decreasing that considerably and then explaining the difference between single-stage object detection vs. other types.
Author Response

(The authors gave the same response as above.)

Round 2
Reviewer 1 Report
I appreciated the authors' reponse and revison.
The revised version is significantly improved. More training details and design details are added. My confusions about some evaluation results are well addressed. Also, the contributions are highlighted, and the following plan is briefly discussed. Overall, the revised version is great. I recommend the paper to be published.